# A Fall Posture Classification and Recognition Method Based on Wavelet Packet Transform and Support Vector Machine

Qingyun Zhang [1], Jin Tao [1,2,*], Qinglin Sun [1], Xianyi Zeng [3], Matthias Dehmer [1,4] and Quan Zhou [2]

1   College of Artificial Intelligence, Nankai University, Tianjin 300350, China; zhangqywell@163.com (Q.Z.); sunql@nankai.edu.cn (Q.S.); Matthias.Dehmer@umit.at (M.D.)
2   Department of Electrical Engineering and Automation, Aalto University, 02150 Espoo, Finland; quan.zhou@aalto.fi
3   Ecole Nationale Supérieure des Arts et Industries Textiles, 59056 Roubaix, France; xianyi.zeng@ensait.fr
4   Department of Computer Science, Swiss Distance University of Applied Sciences, CH-3900 Brig, Switzerland
*   Correspondence: taoj@nankai.edu.cn

**Abstract:** An accidental fall seriously threatens the health and safety of the elderly. The injuries caused by a fall have a lot to do with the different postures during the fall. Therefore, recognizing the posture of falling is essential for the rescue and care of the elderly. In this paper, a novel method was proposed to improve the classification and recognition accuracy of fall postures. Firstly, the wavelet packet transform was used to extract multiple features from sample data. Secondly, random forest was used to evaluate the importance of the extracted features and obtain effective features through screening. Finally, the support vector machine classifier based on the linear kernel function was used to realize the falling posture recognition. The experiment results on "Simulated Falls and Daily Living Activities Data Set" show that the proposed method can distinguish different types of fall postures and achieve 99% classification accuracy.

**Keywords:** falling posture; classification; recognition; wavelet packet transform; support vector machine; random forest

## 1. Introduction

As the number of elderly people in the world continues to rise and the problem of population aging becomes more and more serious, falls are increasingly threatening the lives and health of the elderly [1–3], and an elderly person falls have become an important part of medical care. The physical functions of the elderly over 65 years old are significantly reduced, and at the same time, their reaction ability and balance ability are very weak, and they can easily fall. Once an elderly person falls, the degree of injury of the elderly may be more serious if no one else finds or rescues them, the old person's injury may be more serious, and they may even fall into a coma, endangering their life. In addition, different fall postures during a fall will cause different impact positions, causing varying degrees of injury to the elderly [4,5]. Therefore, there is a need for an automatic detection method, which cannot only recognize the fall behavior of the elderly in time, but also recognize the fall posture, so that the nursing staff can grasp the specific details such as the impact position and the degree of injury of the elderly, and provide the elderly with more targeted rescue and treatment.

In recent years, artificial intelligence technology is developing rapidly. Its application in fall detection has become more and more extensive. Many experts and scholars carried out research work on fall detection and posture recognition. According to the different fall detection equipment, the current fall detection technology can be divided into video, surrounding environment, and wearable fall detection. The method based on video image collects human motion images by deploying one or more video cameras in the user's active area and analyzes the real-time collected video images based on the characteristics of the

fall motion image extracted offline to determine whether the current user has fallen or recognize the posture of a fall [6–11]. The method based on the surrounding environment uses infrared, pressure, and audio sensors arranged in the surrounding environment (walls, floors, ceilings, carpets) to obtain data, analyze, and recognize human behavior [12,13]. The above two methods have the advantages of high accuracy and no need to wear detection equipment, but the whole system is very complicated, incurring high costs and a large amount of data calculation, which easily exposes personal privacy. Moreover, outdoor monitoring cannot be achieved.

With the help of commonly used mobile smart terminals and wearable devices, methods based on wearable devices are widely used to recognize fall postures. Their advantages in terms of accuracy and real-time recognition have made them a research hotspot in the field of healthcare. In this method, a microsensor is worn on the human body to collect data, and a back-end algorithm is used to recognize the falling postures. Most of the studies used acceleration or gyroscope data time series, statistical domain or transform domain features, and used curve similarity comparison or pattern recognition algorithms for fall detection [14–18]. A small set of two or three uniaxial accelerometers mounted on the body were first utilized to distinguish several static and dynamic activities (standing, sitting, lying, walking, ascending stairs, descending stairs, cycling) [19]. He et al. [20] proposed to install an acceleration sensor device on the limbs and crotch and use the support vector machine (SVM) algorithm to judge a variety of typical actions such as jumping, standing still, walking, running, which achieves a detection effect of 92.25% accuracy. Song et al. [21] proposed a method of using a mobile phone as a carrier and embedding a three-dimensional acceleration sensor into it to monitor the body's daily motion posture, covering running, sitting, rising, falling, and other actions, with an accuracy rate of 97.7%. Thanh et al. [22] proposed developing a real-time, simple and high-accuracy fall detection system for the elderly using 3-DOF accelerometers, for which the fall detection algorithm compares the acceleration with the lower fall threshold and upper fall threshold values to accurately detect a fall event; in addition, a post-fall recognition module was added to the method to enhance the performance and accuracy, and the system achieves 100% sensitivity and accuracy. Nevertheless, our research focused on identifying different fall postures, and the threshold method can easily distinguish falls from daily behaviors, but it cannot effectively identify different fall postures. This fall detection method protects users' privacy and has small restrictions on the scope of applications, which is an ideal fall detection technology.

Various sensors on wearable devices provide essential data for fall posture detection. However, the high-dimensional nonlinear data collected by multiple sensors is difficult to process and classify. Therefore, the first but most important thing that needs to be done is to extract features from massive sample data. Most researchers' feature extraction methods mainly include time domain analysis, frequency domain analysis, and time-frequency domain analysis. The time domain and frequency domain analysis methods mainly extract eigenvalues from the signal's time and frequency domain information. Commonly used eigenvalues mainly include mean, standard deviation, variance, root mean square, and peak-to-peak values. For example, Jian et al. [23] used the root mean square of acceleration and angular velocity for fall detection. Tu et al. [24] extracted feature values such as the mean, variance, maximum, and minimum to train a fall detection model. The above methods had a shorter calculation time and lower space complexity, but it cannot fully express the original signal's information and cannot handle non-stationary signals well. The most commonly used time-frequency analysis method was the wavelet transform, which was suitable for analyzing non-stationary signals. Wavelet transform only further decomposes the low-frequency part of the signal and does not decompose the high-frequency part. Compared with the wavelet transform, the wavelet packet transform (WPT) can analyze the signal more finely and decompose the signal's high-frequency part, which is widely used in feature extraction [25–28]. Therefore, the wavelet packet transform can be applied to fall posture recognition to improve the entire system's recognition effect.

Moreover, fall postures can be classified using pattern recognition algorithms, which are good at digging out characteristic parameters that can distinguish different behaviors. Traditional classifiers such as artificial neural networks (ANNs), K-nearest neighbor (KNN) all require many samples to achieve good classification results, and their computational cost is high. At the same time, the processor in the wearable device cannot provide enough memory. However, for fall posture recognition cases, the number of fallen samples is limited, making it challenging to achieve a high recognition accuracy. The support vector machine (SVM) algorithm has excellent advantages in solving small sample and local extreme value problems. Even if the number of samples is limited, it can also achieve good classification results. Compared with ANN, SVM has a low computational cost and small memory occupation and is easy to implement in wearable devices. Aziz et al. [29] presents the research and simulation of wearable device-based fall detection approach by addressing the building of wearable device-based fall detection system for elderly care by using mobile devices. The findings suggest that SVM with the polynomial (order 5) method, which achieved 68.91% overall accuracy and produced only 24.46% FPR, is the most precise model for the fall detection system. Shibuya N. et al. [30] presented a custom-designed wireless gait analysis sensor system for real-time fall detection using an SVM classifier, and six features were extracted for fall classification. Finally, the system achieved an overall specificity of 99.5% and overall sensitivity of 97.0%.

This paper proposes a novel method for fall falling posture classification and recognition based on WPT and SVM and has the following contributions:

1. This paper proposes a fall posture classification and recognition method based on time series analysis and uses SVM to distinguish between various fall postures and daily behaviors, achieving 99% classification accuracy, and good real-time performance;
2. WPT was used to extract multiple features from sample data, and random forest was used to evaluate the importance of the extracted features and obtain effective features through screening. Combining the two can ensure a high accuracy rate of fall gesture recognition, high computational efficiency, and a good application value;
3. The "Simulated Falls and Daily Living Activities Data Set" was chosen to verify the effectiveness of the proposed algorithm, which comes from the UCI database, and is a commonly used standard test dataset for machine learning proposed by the University of California Irvine. The experiment and simulation results show that the proposed method has good real-time performance and high accuracy.

The remainder of this paper is organized as follows. In Section 2, we describe the methods of preprocessing, feature extraction, and classification. In Section 3, we introduce the raw data and the hardware platform, and two sets of experiments are presented to verify the effectiveness of the proposed method. Then, we discuss the limitation of our method and future work. We conclude this paper in Section 4.

## 2. Method

### 2.1. Data Preprocessing

Since a fall action generally lasts 1–3 s, and the original data collection's length is about 15 s, the original data must be intercepted to collect valid data information. The sampling frequency is 25 Hz. Thus, a sliding window with a length of 75 is selected. That is, the detection time is 3 s. In order to intercept effective information, the Algorithm 1 is adopted.

---

**Algorithm 1:** Signal Window Extraction Algorithm

---

```
1: for signal in signals://Each group of data has 21 sensor signals
2:     index = getBiggestChangePoint(signal)
3:     array = arrayAppend(index)
4: index = getMode(array)
5: windowSignal = signals[index-(windowSize/2), index + (windowSize/2)]
```

---

Take the x axis acceleration signal as an example, and the signal before and after the window interception ass shown in Figure 1.

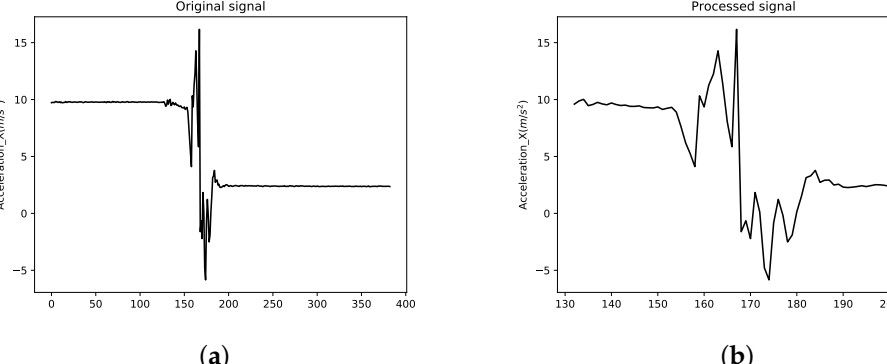

**Figure 1.** Comparison before and after window interception: (**a**) original signal; (**b**) processed signal.

As shown in Figure 1, in the original signal with a length of 15 s, only the data within a period of about 5–8 s are valid data. Using our window interception algorithm, effective data were extracted, which is very important for subsequent feature extraction.

### 2.2. Feature Extraction

Since people's activities of daily living (ADLs) and falls are a dynamic process, the data collected by sensors on wearable devices are usually non-stationary signals. The wavelet packet decomposition can orthogonalize the collected signals in any independent frequency band without omission. The information components contained in the sensor signal are not the same in different fall postures. After wavelet packet decomposition, different frequency bands' energy values can be used as feature variables to express a specific fall posture or action. Therefore, the time-domain features and the features obtained by WPT can be combined as feature variables.

#### 2.2.1. Extract Wavelet Packet Transform Features

We used wavelet packet decomposition to divide the signal into different frequency bands and measure the energy of each frequency band as a feature vector. When performing n-layer wave packet transformation, we chose *db*3 wavelet. Each column of data can obtain $2^n$ groups of different frequency components, and the energy of each frequency component is calculated as follows:

$$e_{n,m} = \sum_{i=1}^{j} |x_{m,i}|^2 \tag{1}$$

where $m = 0, 1, 2 \ldots 2^n - 1, i = 1, 2 \ldots j$, $j$ is the number of discrete sampling points of the signal and $x_{m,i}$ is the amplitude of the discrete points. The decomposition process is shown in the Figure 2.

A certain column of timing signals can be expressed as

$$E_i = [e_{n,0}, e_{n,1}, e_{n,2} \ldots e_{n,m}] \tag{2}$$

Then, all timing signals can be expressed as

$$E = [E_1, E_2 \ldots E_s] \tag{3}$$

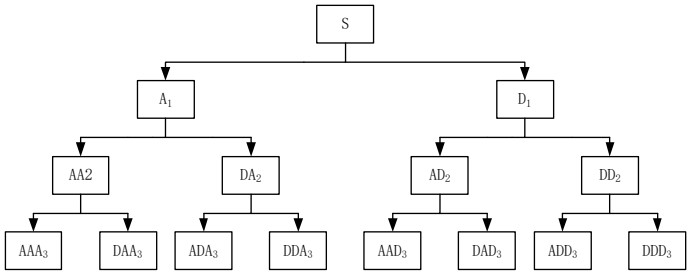

**Figure 2.** Wavelet packet decomposition.

### 2.2.2. Extract Time Domain Features

Feature extraction based on statistical methods is the most commonly used to extract statistical features from time-series signals as feature vectors. This type of feature is suitable for time series signals with prominent statistical characteristics of signal waveforms. For the window signal after preprocessing, we extract certain features with good discrimination from the perspective of statistical methods, and the extracted features are shown in Table 1:

**Table 1.** Several time-domain features extracted.

| Time-Domain Features | Expression |
| --- | --- |
| Root mean square (RMS) | $RMS(x) = \sqrt{\frac{\sum_{i=1}^{n} x_i^2}{n}}$ |
| Peak-to-peak (pk-pk) | $pk - pk(x) = \max(x) - \min(x)$ |
| Peak factor (pf) | $pf(x) = \max(x)/RMS(x)$ |
| Waveform factor (wf) | $wf(x) = RMS(x)\frac{n}{\sum_{i=1}^{n}|x_i|}$ |
| Pulse factor (pf) | $pf(x) = \max(x)\frac{n}{\sum_{i=1}^{n}|x_i|}$ |
| Margin factor (mf) | $mf(x) = \max(x)\left(\frac{n}{\sum_{i=1}^{n}x_i}\right)^2$ |
| Kurtosis (K) | $K(x) = \frac{\frac{1}{n}\sum_{i=1}^{n}x_i^4}{\left(\frac{1}{n}\sum_{i=1}^{n}x_i^2\right)^2}$ |

### 2.3. Feature Selection

Through feature extraction, hundreds of features can be obtained, but not all features have a positive effect on model classification and recognition. Therefore, it is necessary to reduce the dimensionality of features to eliminate irrelevant features.

Random forest is an algorithm that integrates multiple trees through the idea of ensemble learning. Its basic unit is a decision tree, and its essence belongs to ensemble learning. Random forest combines hundreds of decision trees, trains each decision tree on slightly different observation sets, and considers only a limited number of features in each tree to split nodes. The final prediction of random forest is through averaging the predictions of each tree. The introduction of two random trees makes mitigates the chances of the random forest falling into over-fitting and provides it with good anti-noise ability. It can evaluate each variable's importance and perform feature selection on high-dimensional data, thereby making the model more robust.

The calculation method of the importance of a certain feature $X$ in a Random Forest is shown as follows:

1. For each decision tree in the Random Forest, we use the corresponding out-of-bag (OOB) data to calculate its error, denoted as $err_{oob}1$;
2. Randomly add noise interference to the feature $X$ of all samples of OOB data (the value of the sample at feature $X$ can be randomly changed), and calculate its out-of-bag data error again and record it as $err_{oob}2$;
3. Assuming that there are $N$ trees in the random forest, then the importance of feature is expressed as $X = \sum(err_{oob}2 - err_{oob}1)/N$. After a certain feature is randomly added

to the noise, the accuracy rate outside the bag is greatly reduced, which means that this feature greatly influences the classification result of the sample.

*2.4. Classification*

2.4.1. Use SVM to Deal with Two Classification Problems

SVM is a generalized linear classifier that classifies binary or multivariate data according to the supervised learning method, and its decision boundary is the maximum margin hyperplane for solving learning samples. The SVM separates different sample categories by constructing the optimal classification plane to maximize the boundary distance between two or more categories. The simplest SVM model is the maximum interval classifier. It is only suitable for the case where the sample data are linearly separable in the feature space. However, in practice, the training sample set is usually linearly inseparable, so parameter relaxation variables $\xi$ are introduced. If the sample is misclassified during classification, $\xi$ will be greater than 0. The following linear equation can describe any hyperplane:

$$\omega^* \cdot x + b^* = 0 \tag{4}$$

The classification decision function is defined as

$$f(x) = sign(\omega^* \cdot x + b^*) = sign\left[\sum_{i=1}^{m} \alpha_i^* y_i(x_i \cdot x) + b^*\right] \tag{5}$$

Let $\sum_{i=1}^{n} \xi_i$ describe the degree of misclassification of training samples. For the classification interface in the case of linear inseparability, it is hoped that the maximum classification interval $2/\|\omega\|$ should be as large as possible, and the degree of misclassification of training samples $\sum_{i=1}^{n} \xi_i$ should be as small as possible. When errors occur, the degree of punishment for errors needs to be controlled, so the penalty factor $C$ is introduced, which is an adjustable parameter. The larger $C$ is, the heavier the penalty is. Consequently, the objective function can be written as:

$$\min\left[\frac{1}{2}\|\omega\|^2 + C\left(\sum_{i=1}^{n} \xi_i\right)\right] \tag{6}$$

$$s.t. y_i(\omega^T x_i + b) \geq 1 - \xi_i, \xi_i \geq 0, i = 1, 2, ..., n$$

Rewritten by Lagrangian duality transformation, we have:

$$\max\left[\sum_{i=1}^{m} \alpha_i - \frac{1}{2}\sum_{i,j=1}^{m} \alpha_i \alpha_j y_i y_j(x_i \cdot x_j)\right] \tag{7}$$

$$s.t. \sum_{i=1}^{m} \alpha_i y_i = 0, 0 \leq \alpha_i \leq C, i = 1, 2, ..., m$$

Since the training samples are linearly inseparable, the original sample can be transformed into a high-dimensional space by introducing a kernel function method using nonlinear mapping. That is, the linear inseparable problem in the low-dimensional space is transformed into the linearly separable problem in the high-dimensional space. Common kernel functions are shown in Table 2:

**Table 2.** Common kernel functions in SVM.

| Kernel Function | Kernel Function Expression | Parameter |
| --- | --- | --- |
| Linear kernel | $K(X_i \cdot X) = (X_i \cdot X)$ | - |
| Polynomial kernel function | $K(X_i \cdot X) = [s(X_i \cdot X) + c]^d$ | $s, c, d$ |
| Radial basis function | $K(X_i \cdot X) = \exp\left(-g\|X - X_i\|^2\right)$ | $g$ |
| Sigmoid kernel function | $K(X_i \cdot X) = \tanh[s(X_i \cdot X) + c]$ | $s, c$ |

Choosing a suitable kernel function, the optimal classification function can be obtained:

$$f(X) = sign\left[\sum_{i=1}^{m} \alpha_i^* y_i K(x_i, x) + b^*\right] \tag{8}$$

### 2.4.2. Multi-Classification Method of SVM

The classic SVM only gives a two-class classification algorithm, so extending the support vector machine to a multi-class classification problem and obtaining significant results is still challenging. When using SVM to construct multi-class classifiers, two methods are usually applied. One is to extend to a multi-class classification support vector machine. This method uses a large number of variables and has high computational complexity, which is not applicable in reality. The other is to convert into a two-class classification problem using several two-class classification support vector machines to form a multi-class classifier. Due to the fast calculation speed, this method is widely used. This method has two typical algorithms called the "one-against-all" algorithm and the "one-against-one" algorithm.

The "one-against-all" algorithm sequentially classifies the samples of a specific category into one category and other remaining samples into another category. In this way, the sample of $N$ categories constructs $N$ second-class classifiers. When classifying, the samples to be tested are classified into the category with the largest classification function value. Although this algorithm has the advantages of a small number of classifiers and a fast classification speed, due to the large difference in sample size between the two types, it is easy to cause the problem of dataset skew. At the same time, the "one-against-one" algorithm uses the training samples of any two categories to train a two-class classifier without repetition. For a multi-class classification problem with $N$ categories of training samples, a total of $N * (N - 1)/2$ second-class classifiers is required. Although the number of classifiers in this algorithm is large, the training scale of a single second-class classifier of this algorithm is small, and the training data are balanced and easy to expand. In this paper, the "one-against-one" algorithm is applied to construct a multi-class classifier.

## 3. Results and Discussions

### 3.1. Experimental Setting

#### 3.1.1. Data Source

The dataset used in this paper comes from the UCI database. The "Simulated Falls and Daily Living Activities Data Set" [31] included in the database in June 2018 was chosen for our experiment. The dataset was completed by 17 volunteers, including ten men and seven women. Each volunteer must complete 36 behaviors, including 20 falling behaviors and 16 daily behaviors, each repeated five times. In this way, there are 3060 sampled data for each part. The original signal is a total of 21-dimensional signals such as three-dimensional acceleration, three-dimensional angular velocity, three-dimensional velocity, three-dimensional geomagnetism, pressure, and more output from the sensor. The sensor's sampling frequency is 25 Hz, and the average sampling time for each action is 15 s. It has been shown that breasts can achieve better recognition accuracy [32]. In this paper, we select all samples of breasts in this dataset as raw data. Figure 3 shows the change of the sensor value during front-knees and back-lying actions. It can be observed that

different sensors produce different signals. Thus, different fall postures can be identified by extracting the change characteristics of different sensors' signal values.

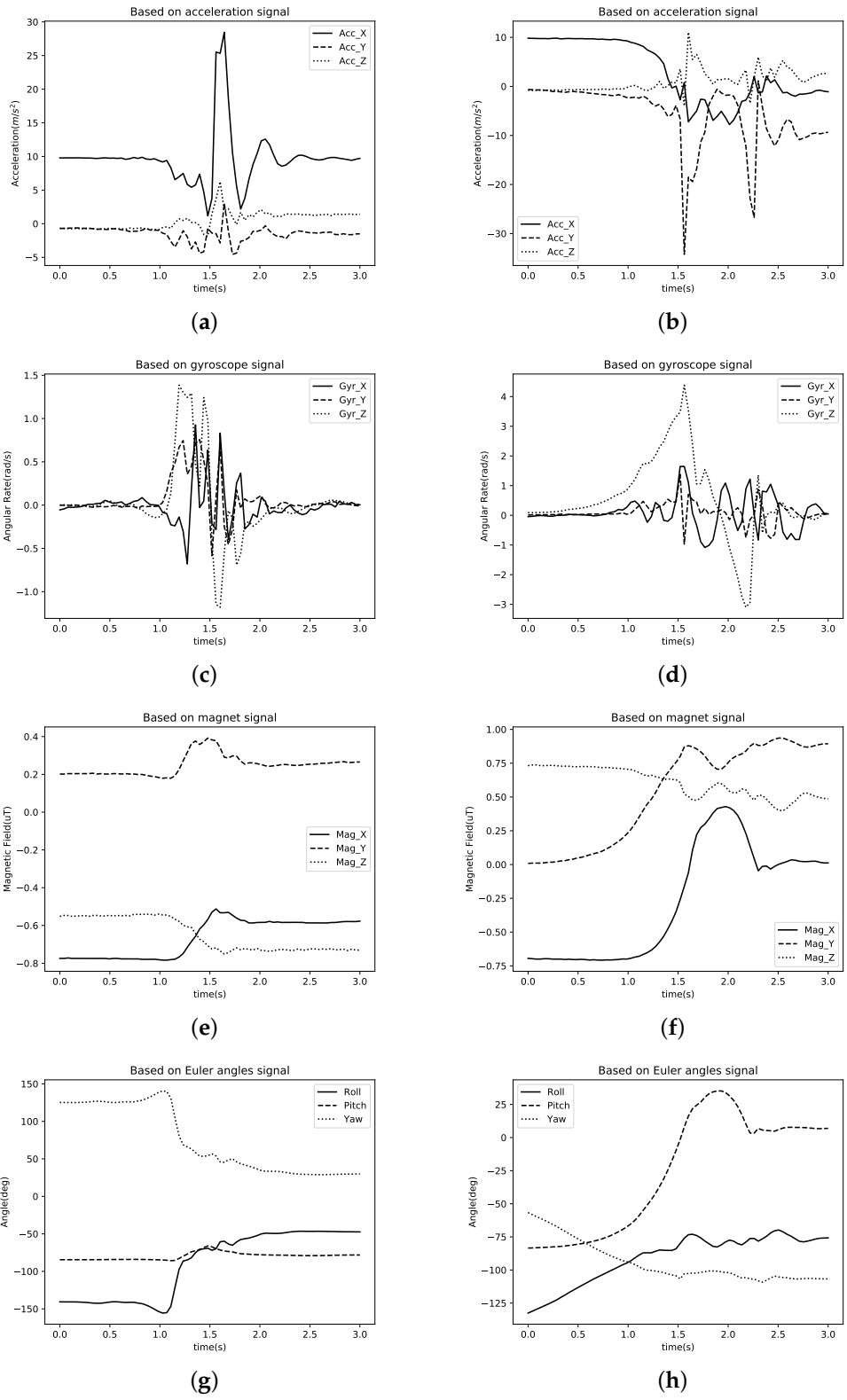

**Figure 3.** Examples of sensor signals for a front-knees action and a back-lying action: (**a**) front-knees, acceleration; (**b**) back-lying, acceleration; (**c**) front-knees, gyroscope; (**d**) back-lying, gyroscope; (**e**) front-knees, magnet; (**f**) back-lying, magnet; (**g**) front-knees, Euler angle; and (**h**) back-lying, Euler angle.

### 3.1.2. Data Division

There are 20 fall behaviors in the original data, of which we selected 12 behaviors. In the unused fall data, the four actions numbered 905, 906, 914, and 916 are those the tester would quickly or slowly recover after a fall, which shows that this type of fall does not cause the user to lose control of their body, so it is beyond the scope of our research. The remaining four behaviors numbered 917, 918, 919, and 920 did not indicate the fall's direction and posture, so these four datasets were discarded. In addition, this paper's focus is to recognize the posture when falling, and the 16 daily behaviors were classified into one category, and no further subdivision was required. Finally, the selected 12 fall behaviors and 16 daily behaviors were classified into seven different behaviors, which were ADLs, front-lying, front-knees, back-lying, back-sitting, fall to the left, fall to the right.

### 3.1.3. Experimental Platform

The research on the falling posture recognition and classification algorithm based on machine learning was completed on the Windows10 operating system; we used the PyCharm 2019.2.2 (Community Edition) as software, and the CPU type was Intel Core i5-4200H @2.80 GHz.

### *3.2. Experimental Results*

Based on the "Simulated Falls and Daily Living Activities Data Set", this paper extracted 12 fall actions and 16 ADLs such as standing, walking, squatting, and a total of 3060 samples to test the proposed fall posture recognition method. It was pointed out that [33] there is no common standard for evaluating the effect of fall posture recognition at this stage, and it is challenging to perform effect testing based on general procedures. In this paper, accuracy, recall, f1 score and precision were used to evaluate the testing effect. There are four basic symbols: true positive (*TP*): successfully predict positive samples as positive, true negative (*TN*): successfully predict negative samples as negative; false positive (*FP*): wrongly predict negative samples as positive; false negative (*FN*): wrongly predict positive samples as negative. For multi-classification problems, the macro method can be used. The accuracy, precision, recall, and f1 score can be expressed as

$$Accuracy = \frac{TP + TN}{TP + TN + FP + FN} \tag{9}$$

$$Precision = \frac{TP}{TP + FP}$$
$$macro - Precision = \frac{1}{n} \sum_{i=1}^{n} Precision_i \tag{10}$$

$$Recall = \frac{TP}{TP + FN}$$
$$macro - Recall = \frac{1}{n} \sum_{i=1}^{n} Recall_i \tag{11}$$

$$F_{1score} = \frac{2 * Precison * Recall}{Precison + Recall}$$
$$macro - F_{1score} = \frac{1}{n} \sum_{i=1}^{n} F_i \tag{12}$$

### 3.2.1. Result 1: Comparison between before and after Feature Extraction of WPT Using Different Classifiers

In order to verify the effectiveness of the proposed wavelet packet decomposition method in feature extraction and the recognition and classification of SVM, comparison between the sample set formed by the time-domain feature vector combined with the feature vector obtained by wavelet packet decomposition with the sample set that only uses the time-domain feature vector were conducted. At the same time, the SVM based on linear kernel function (Linear_SVM), SVM based on radial basis function (Rbf_SVM), Xgboost, LogisticRegression and KNN are used for the recognition and classification of fall posture. The classification of the selected 16 ADLs and 12 fall behaviors are shown in Table 3. In order to better evaluate the prediction performance of the model, the experimental results all adopted a five-fold cross-validation method, and finally, the fall posture recognition results of different classifiers are shown in Table 4. We observed that, regardless of whether we used the time-domain feature or the wavelet packet transform feature combined with the time-domain feature, the accuracy, recall, f1 score, and precision of the SVM with the linear kernel function reached the highest. In addition, we can see that for each classifier, after adding the features obtained by the wavelet packet transform, the accuracy, recall, f1 score, and precision were greatly improved. In order to test the significance of the differences as mentioned above, this paper used the multi-factor analysis of variance to test the significance of the differences in the experimental results using wavelet packet transform and different classifiers, for which the significance level $\alpha$ is set to 0.05; if the p-value was less than the $\alpha$, it means that there is a statistically significant difference in the above results. The experimental results are shown in Table 5. The results indicate that SVM has significant advantages in solving the problem of small samples and high-dimensional features, and it also confirms the effectiveness of the wavelet packet transform in feature extraction.

**Table 3.** Classification of the selected 16 ADLs and 12 fall behaviors.

| Number | Behavior Description | Type |
|:---:|:---:|:---:|
| 801 | walking forward | |
| 802 | walking backward | |
| 803 | jogging,running | |
| 804 | squatting, then standing up | |
| 805 | bending about 90 degrees | |
| 806 | bending to pick up an object on the floor | |
| 807 | walking with a limp | |
| 808 | stumbling with recovery | |
| 809 | bending while walking and then continuing walking | ADLs |
| 810 | coughing or sneezing | |
| 811 | to sitting with a certain acceleration onto a chair (hard surface) | |
| 812 | to sitting with a certain acceleration onto a sofa (soft surface) | |
| 813 | to sitting in the air exploiting the muscles of legs | |
| 814 | to sitting with a certain acceleration onto a bed (soft surface) | |
| 815 | from vertical lying on the bed | |
| 816 | from lying to sitting | |
| 901 | from vertical falling forward to the floor | |
| 902 | from vertical falling forward to the floor with arm protection | front-lying |
| 907 | from vertical falling down on the floor, ending in right lateral position | |
| 908 | from vertical falling down on the floor, ending in left lateral position | |

**Table 3.** *Cont.*

| Number | Behavior Description | Type |
|---|---|---|
| 903 | from vertical falling down on the knees | front-knees |
| 904 | from vertical falling down on the knees and then lying on the floor | |
| 910 | from vertical falling on the floor, ending lying | back-lying |
| 911 | from vertical falling on the floor, ending lying in right lateral position | |
| 912 | from vertical falling on the floor, ending lying in left lateral position | |
| 909 | from vertical falling on the floor, ending sitting | back-sitting |
| 915 | from vertical falling on the floor, ending lying | fall to the left |
| 913 | from vertical falling on the floor, ending lying | fall to the right |

**Table 4.** Comparison of different methods: accuracy, recall, f1 score and precision on different classifiers with or without WPT.

| | | Linear_SVM | Rbf_SVM | Xgboost | Logistic Regression | KNN |
|---|---|---|---|---|---|---|
| Accuracy | without WPT | 0.967 | 0.939 | 0.936 | 0.941 | 0.934 |
| | with WPT | 0.987 | 0.974 | 0.978 | 0.973 | 0.965 |
| Recall | without WPT | 0.955 | 0.932 | 0.930 | 0.936 | 0.902 |
| | with WPT | 0.973 | 0.938 | 0.952 | 0.940 | 0.935 |
| f1 score | without WPT | 0.954 | 0.951 | 0.939 | 0.940 | 0.913 |
| | with WPT | 0.974 | 0.955 | 0.960 | 0.945 | 0.933 |
| Precision | without WPT | 0.953 | 0.973 | 0.949 | 0.947 | 0.927 |
| | with WPT | 0.977 | 0.975 | 0.967 | 0.952 | 0.935 |

**Table 5.** Significance test of experimental results.

| | $p$-Value (Use WPT) | Significant? | $p$-Value (Use Different Classifiers) | Significant? |
|---|---|---|---|---|
| Accuracy | 0.000851 | Yes | 0.045815 | Yes |
| Recall | 0.036004 | Yes | 0.038582 | Yes |
| f1 score | 0.022702 | Yes | 0.016202 | Yes |
| Precision | 0.047582 | Yes | 0.014486 | Yes |

3.2.2. Result 2: Comparison before and after Feature Selection

Combining the extracted time-domain features and wavelet packet features as a new sample set can further improve the classifier's recognition accuracy. However, the feature dimension is too high. Only certain features can help the classifier perform classification and recognition. The remaining features will affect the recognition accuracy and cause a tremendous waste of computing resources and time. Table 6 compares the recognition accuracy, recall, f1 score, precision, training time, and real-time decision time of the algorithm before and after the random forest feature selection. The classifier used is a support vector machine with a linear kernel function. We can observe that after feature selection through the random forest, the algorithm's accuracy, recall, f1 score and precision are all further improved, and the training time and real-time decision time are also significantly shortened.

**Table 6.** Comparison before and after feature selection based on Linear_SVM.

|  | Before Feature Selection | After Feature Selection |
|---|---|---|
| Accuracy | 0.986504 | 0.990553 |
| Recall | 0.972577 | 0.978491 |
| f1 score | 0.974415 | 0.979512 |
| Precision | 0.976582 | 0.981099 |
| Training time | 264.93 (s) | 7.59 (s) |
| Real-time decision time | 4.29 (s) | 0.12 (s) |

### 3.2.3. Comparisons with Existing Methods

In addition, in order to verify the effectiveness and superiority of the method in this paper, comparisons were made with several existing excellent methods on "Simulated Falls and Daily Living Activities Data Set". Results are as shown in Table 7. The experimental results show that the accuracy of the method proposed in this paper is better than that of the other three methods.

**Table 7.** Comparison of different methods.

| Method | Accuracy |
|---|---|
| Tong [34] based on one-class SVM | 0.932 |
| Zhou [35] based on BP neural network | 0.901 |
| Hou [36] based on SVM | 0.922 |
| Proposed in this paper | 0.991 |

### *3.3. Discussions*

### 3.3.1. Discussion of the Limitation

Although the method proposed in this paper can effectively recognize the posture of a fall, it requires the wearing of many sensors. We need to consider the comfort of wearing these sensors for the elderly. In addition, the method proposed in this paper does not perform learning, training, and recognition for some special fall postures, such as from lying down, rolling out of bed, and walking to the floor, falling down from standing, and slowly slipping on a wall. Therefore, these special behaviors may not be effectively recognized.

### 3.3.2. Discussion on the Future Work

Although the method proposed in this paper has been simulated on a computer and achieved good results, it still needs to be verified in practical applications. Then, we will embed the model trained on the computer into the mobile phone app, use wearable sensors to collect data, and send the data to the mobile phone via Bluetooth. The mobile phone processes the data in real time, determines whether the elderly fell, and recognizes the posture of the fall. Once the elderly has fallen, the mobile phone will immediately send the posture and location information of the fall to family members or medical staff so that the elderly can be rescued in time.

### 4. Conclusions

This paper proposed a fall posture classification and recognition method based on WPT and SVM. To be specific, the algorithm uses the WPT to extract the original signal features, random forest for feature selection, and SVM for classification. The following conclusions can be drawn:

1.  The energy ratio of the frequency band obtained after wavelet packet decomposition can be used as features to show the characteristics of the original signal and improve the classifier's recognition effect;
2.  Using random forest for feature screening and constructing a new feature set can further improve the accuracy of recognition and shorten the algorithm's training time and real-time decision time;

3.  Compared with other classifiers, by using the linear kernel function the SVM can effectively improve the accuracy of classification, and the recognition effect is the best.

**Author Contributions:** Q.Z. (Qingyun Zhang): methodology, software, writing—original draft, writing—review and editing. J.T.: conceptualization, writing—review and editing. Q.S.: review and editing, supervision. X.Z., M.D., Q.Z. (Quan Zhou): review and editing, supervision. All authors have read and agreed to the published version of the manuscript.

**Funding:** This work was supported by the National Natural Science Foundation of China (Grant No. 61973172, 61973175, 62003175 and 62003177), the National Key Research and Development Project (Grant No. 2019YFC1510900), the key Technologies Research and Development Program of Tianjin (Grant No. 19JCZDJC32800), this project also funded by China Postdoctoral Science Foundation (Grant No. 2020M670633) and Academy of Finland under No. 315660.

**Informed Consent Statement:** Informed consent was obtained from all subjects involved in the study.

**Data Availability Statement:** The data presented in this study are openly available in UCI Machine Learning Repository.

**Conflicts of Interest:** The authors declare no conflict of interest.

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
