# Peer review of "A Fall Posture Classification and Recognition Method Based on Wavelet Packet Transform and Support Vector Machine"

_applsci, doi:10.3390/app11115030_

Round 1

Reviewer 1 Report

The article presents the use of Wavelet Packet Transform (WPT) for extracting relevant signals from wearable devices and its use for detecting different types of falls in humans, which can be critical to assess the cause of the fall and acting in consequence. Apart from that, a feature selection with Random Forests is proposed. The authors propose the use of Support Vector Machines to classify the signals in the different classes (no fall and six different fall positions) and compare the results with other classifiers. The results evaluate the influence of the WPT processing in several  classifiers and the effect of the feature selection in an SVM classifier with linear kernel. Results show that a linear SVM with WPT is better than other options and that feature selection provides slightly improvements in classification and high reductions of the processing time.

The paper is in general good. The presentation of the motivations, the models and the datasets are quite comprehensive and clear. I have a few concerns that must be solved to obtain a final publishable version.

My major concern is on the results. First, presenting the results with different classifiers with and without WPT would be better presented in tables instead of bar graphs. Moreover, given that you have only 3600 test samples, with 3 decimals would be enough to present all the results. Related to that, since the number of test samples is not very high and the differences are small, you should perform statistical significance tests in order to assess if those differences are statistically significant. This would allow to support your claim on that SVM are the best option for classifying, apart from corroborate the real impact of WPT and Random Forest feature selection.

Apart from that, there are some small problems to solve:

  • In the first paragraph of the introduction you state the relation between the fall position and the consecuences; it would be necessary to provide a bibliographic reference in the medical domain that confirms these statements
  • - Section 2.2.1 is written in a very strange way, it seems like a sequence of orders instead of a description
  • Figure 2 is never referenced in the text
  • The first time you write "Random Forest" you do with capitals, but in the rest of the text you do not follow the same criteria; you should unify the case
  • In Eq. (8) you are using X (capital), while in the previous equations you use it lowercase (x); is there any reason for that? If not, unify the case
  • The description of the "one-against-all" in lines 197 and 198 is not clear; please re-write it in a clearer manner
  • In Equations (10), (11) and (12) please separate the macro definitions from their counterparts
  • There is no future work; are you not continuing research in this field?

Finally, please proofread the paper becase there are several English mistakes (innappropriate use of some words, missing articles, etc.) that should be corrected.

Author Response

Reviewer #1,Concern # 1: Presenting the results with different classifiers with and without WPT would be better presented in tables instead of bar graphs. Moreover, given that you have only 3600 test samples, with 3 decimals would be enough to present all the results. Related to that, since the number of test samples is not very high and the differences are small, you should perform statistical significance tests in order to assess if those differences are statistically significant.

Author response: Thank you very much for raise this concern. According to your suggestion, we have displayed the results of experiment 1 in the form of a table, and the results of the experiment are kept to 3 decimal places, as shown in Table 4. In fact, due to the insufficient number of samples in the data set, our experimental results all adopt 5-fold cross-validation, but it is not enough. We used the multi-factor analysis of variance to test the significance of the differences, as shown in table 5. However, because the data in Table 6 has only two data before and after feature selection for each evaluation index, such as accuracy, recall, etc., we cannot calculate the P-value. Therefore we only compare the size of the two numbers before and after feature selection. We added a description of statistical significance tests and updated the table on Page 10 and Page 11.

Reviewer #1, Concern # 2: In the first paragraph of the introduction you state the relation between the fall position and the consecuences; it would be necessary to provide a bibliographic reference in the medical domain that confirms these statements 

Author response: Thank you for bringing this concern. We have added references [4,5] to explain the relation between the fall position and the consequences. Different postures and impact positions when falling can cause varying degrees of damage to the elderly. We have added a description of the relation on the first paragraph of the introduction.

Reviewer #1, Concern # 3: Section 2.2.1 is written in a very strange way, it seems like a sequence of orders instead of a description.

Author response: Thank you so much for pointing this out. We have reorganized the language to illustrate the process of extracting wavelet packet transform features on page 4 .

Reviewer #1, Concern # 4: Figure 2 is never referenced in the text. The first time you write "Random Forest" you do with capitals, but in the rest of the text you do not follow the same criteria; you should unify the case In Eq. (8) you are using X (capital), while in the previous equations you use it lowercase (x); is there any reason for that? If not, unify the case.

Author response: Thank you so much for pointing this out. We are sorry for all the mistakes caused by carelessness . We have gone through the whole paper carefully and tried our best to correct the errors you pointed out. Also, we quoted Figure 2 on page 4, unified the case X in equations. 

Reviewer#1, Concern # 5: The description of the "one-against-all" in lines 197 and 198 is not clear; please re-write it in a clearer manner. In Equations (10), (11) and (12) please separate the macro definition s from their counterparts. There is no future work; are you not continuing research in this field? Finally, please proofread the paper because there are several English mistakes (inappropriate use of some words, missing articles, etc.) that should be corrected.

Author response: Thank you so much for bringing this concern. We have re-described the concept of the "one-against-all" on page 7. We have separated the macro definitions from their counterparts in Equations (10), (11) and (12). In addition, we have added section 3.3.2 to discuss future work. And finally, we corrected some English grammar errors.

Reviewer 2 Report

In this paper, the authors proposed a method to improve the classification and recognition accuracy of fall postures.
+ The authors presented the comparison between before and after feature extraction of WPT using different classifiers; Comparison before and after feature selection. However, it is not enough. A Discussion section should be added to this paper. In this section, sufficient comparisons with the previous works must be added.
+ To have an unbiased view of the manuscript, there should be some discussions on the limitations of the proposed method.
+ The authors did not make a real device to collect the data and detect the fall event in a real-time manner. However, the authors should present how to feed their algorithm into an embedded device.

Author Response

Reviewer #2, Concern # 1: The authors presented the comparison between before and after feature extraction of WPT using different classifiers; Comparison before and after feature selection. However, it is not enough. A Discussion section should be added to this paper. In this section, sufficient comparisons with the previous works must be added.

Author response: Thank you for bringing this concern. We realized that the comparison between before and after feature extraction of WPT using different classifiers and comparison before and after feature selection are not enough, so we add section 3.2.3. In this section, we compared several existing excellent methods with our methods, the results are as shown in table 7.

Reviewer #2, Concern # 2: To have an unbiased view of the manuscript, there should be some discussions on the limitations of the proposed method.

Author response: Thank you so much for bring this concern. In fact, our method requires the use of many wearable sensors, so we must consider the issue of wearing comfort for the elderly. In addition, due to the problem of insufficient experimental samples, our method cannot recognize some special falling postures. We have added discussion in Section 3.3.1.  

Reviewer #2, Concern # 3: The authors did not make a real device to collect the data and detect the fall event in a real-time manner. However, the authors should present how to feed their algorithm into an embedded device.

Author response: Thank you so much for your comment. In fact, we have already considered how to apply our method to actual experiments, we have embedded our algorithm into a mobile app to recognize of the fall posture of the elderly. However, We have added mentioned points in section 3.3.2 as future work.

Round 2

Reviewer 2 Report

1/ Please discuss the accuracy concerned in the following study because, in this study, the performance index is very good.

Development of a Real-time, Simple and High-Accurate Fall Detection System for Elderly Using 3-DOF Accelerometers, Arabian Journal for Science and Engineering (AJSE), ISSN 1319-8025, SPRINGER,44, pages3329–3342(2019) https://doi.org/10.1007/s13369-018-3496-4, 2018.

2/ The reviewer suggests that the editor may accept this paper with the above minor concern.

Author Response

Reviewer #2,Concern # 1: Please discuss the accuracy concerned in the following study because, in this study, the performance index is very good.

Development of a Real-time, Simple and High-Accurate Fall Detection System for Elderly Using 3-DOF Accelerometers, Arabian Journal for Science and Engineering (AJSE), ISSN 1319-8025, SPRINGER,44, pages3329–3342(2019) https://doi.org/10.1007/s13369-018-3496-4, 2018.

Author response: Thank you very much for your comment. We thank you for providing us such a good reference. We have carefully studied the paper you provided. As far as we know, this paper proposes an algorithm based on the threshold method. Although the method is simple, the recognition accuracy is very high, the calculation speed is very fast, and it does not consume too much hardware resources. We would like to apply this algorithm in our study. However, our research in this paper focuses on identifying different fall postures. The threshold method can easily distinguish falls from daily behaviors, but it cannot effectively identify different fall postures. In order to more accurately identify different fall postures, it is necessary to extract effective features from the sensor signals. In the revised version of the manuscript, we have discussed the mentioned paper in related work.
